# Antibacterial and Antibiofilm Activity of Chemically and Biologically Synthesized Silver Nanoparticles

**DOI:** 10.3390/antibiotics12071084

**Published:** 2023-06-21

**Authors:** Karen Verduzco-Chavira, Alba Adriana Vallejo-Cardona, Angélica Sofía González-Garibay, Omar Ricardo Torres-González, Iván Moisés Sánchez-Hernández, Jose Miguel Flores-Fernández, Eduardo Padilla-Camberos

**Affiliations:** 1Department of Technological and Industrial Processes, ITESO, The Jesuit University of Guadalajara, Anillo Perif. Sur Manuel Gómez Morin 3838, Tlaquepaque 45604, Mexico; ib720653@iteso.mx; 2Medical and Pharmaceutical Biotechnology Unit, Center for Research and Assistance in Technology and Design of the State of Jalisco, A.C. (CIATEJ), Av. Normalistas No. 800 Col. Colinas de la Normal, Guadalajara 44270, Mexico; avallejo@ciatej.mx (A.A.V.-C.); nutricion.integral@yahoo.com (A.S.G.-G.); polimerasados@gmail.com (O.R.T.-G.); iv_23_36@hotmail.com (I.M.S.-H.); 3Department of Biochemistry & Centre for Prions and Protein Folding Diseases, University of Alberta, 204 Brain and Aging Research Building, Edmonton, AB T6G 2M8, Canada; floresfe@ualberta.ca; 4Departamento de Investigación e Innovación, Universidad Tecnológica de Oriental, Oriental C.P., Puebla 75020, Mexico

**Keywords:** silver nanoparticles, chemical synthesis, biological synthesis, antibiofilm, *Jacaranda mimosifolia*

## Abstract

Bacterial biofilms are a significant problem in the food industry, as they are difficult to eradicate and represent a threat to consumer health. Currently, nanoparticles as an alternative to traditional chemical disinfectants have garnered much attention due to their broad-spectrum antibacterial activity and low toxicity. In this study, silver nanoparticles (AgNPs) were synthesized by a biological method using a *Jacaranda mimosifolia* flower aqueous extract and by a chemical method, and the factors affecting both syntheses were optimized. The nanoparticles were characterized by Ultraviolet–visible (UV–Vis) spectrophotometry, Fourier-transform infrared spectroscopy (FTIR), Dynamic light scattering (DLS), X-ray diffraction (XRD), and Transmission electron microscopy (TEM) with a spherical and uniform shape. The antibacterial and antibiofilm formation activity was carried out on bacterial species of *Pseudomonas aeruginosa* and *Staphylococcus aureus* with the capacity to form biofilm. The minimum inhibitory concentration was 117.5 μg/mL for the chemical and 5.3 μg/mL for the biological nanoparticles. Both types of nanoparticles showed antibiofilm activity in the qualitative Congo red test and in the quantitative microplate test. Antibiofilm activity tests on fresh lettuce showed that biological nanoparticles decreased the population of *S. aureus* and *P. aeruginosa* by 0.63 and 2.38 logarithms, respectively, while chemical nanoparticles had little microbial reduction. In conclusion, the biologically synthesized nanoparticles showed greater antibiofilm activity. Therefore, these results suggest their potential application in the formulation of sanitizing products for the food and healthcare industries.

## 1. Introduction

Biofilms are complex microbial communities made up of one or several species embedded in a self-produced extracellular matrix, which is mainly composed of polysaccharides, proteins, and exogenous DNA [1,2]. This matrix acts as a mucus-like protective barrier that allows microorganisms to adhere to surfaces, resist environmental stresses, and evade the host’s immune response. As a key virulence factor for pathogenicity, biofilm also provides protection against disinfectants and treatments for their elimination [3].

The formation of biofilms causes undesirable effects in both medical and non-medical areas. In non-medical areas, biofilms can lead to the formation of biofouling on objects or surfaces that remain in contact with seawater, such as ship hulls and underwater gas and oil transport pipes, as well as in water distribution systems [4,5]. In the medical field, biofilms are a major public health problem, causing chronic infections in humans that, if not treated properly, can become complicated and persistent. They can also be found in hospital facilities and medical devices [6].

In the food industry, biofilms can develop in different places, including water and other liquid pipes, reverse osmosis membranes, pasteurization plates, and, tables, employees’ gloves, and any food contact or non-food contact surfaces [7,8,9], regardless of the material, whether it is stainless steel, wood, glass, rubber, or propylene, where raw materials and additives are stored. Even raw materials, such as meat, fish, bones, vegetables, and fruits, can be affected by biofilm formation [1].

The formation of biofilms in the food industry results in changes in the sensory properties of food and a decrease in work efficiency due to issues with equipment and surfaces where bacteria can develop and spread. The presence of biofilms represents a potential health risk to the population due to the consumption of food with biofilms of microorganisms, such as *Bacillus cereus*, *Escherichia coli*, *Listeria monocytogenes*, *Salmonella enterica*, *Staphylococcus aureus*, and *Pseudomonas aeruginosa*, which can cause diseases [1,2,10]. In the United States alone, between 1990 and 2005, there were over 30,000 cases of diseases associated with the consumption of fresh products [11], and 80% of these cases were attributed to biofilm-forming microorganisms [12].

Chemical disinfectants have been the primary choice for combating biofilm formation in food industries. However, these treatments have their limitations. Biofilm’s protective nature makes it challenging to eradicate effectively, leading to the generation of toxic byproducts [13]. Moreover, disinfectants can become less effective over time, as microorganisms develop resistance to them. Additionally, some disinfectants, including quaternary ammonium compounds, can pose potential risks to human health and the environment, as this compound has been associated with developing respiratory diseases, such as asthma. These challenges emphasize the importance of finding alternative approaches to combat biofilms in the food industry that are both efficient and safe for the environment and human health. The use of silver nanoparticles (AgNPs) has recently been proposed as a promising alternative to combat biofilms due to their unique properties, such as large surface area, which can be in contact with microorganisms; high reactivity; and the possible ability to penetrate the biofilm matrix. Some studies have demonstrated the antibiofilm effect of AgNPs synthesized by different methods [14,15,16].

There are many ways to synthesize AgNPs. These can be divided into chemical, physical, and biological synthesis. The chemical methods use solvents that, although they guarantee a stable size and shape, generate a lot of pollution in the environment; on the other hand, the physical methods, although they do not generate as much environmental impact, do require a greater amount of energy. That is why biological synthesis has been proposed as an alternative, since different biological agents, such as fungi, bacteria, and plant extracts, can be used, the last being easy to access, since they are easy to obtain and are not expensive nor harmful to the environment; in addition, since there is a wide variety of silver-reducing compounds in plants, they offer advantages over nanoparticles synthesized with other methods [17,18,19,20].

These advantages include that they are eco-friendly because the biosynthesis process tends to use fewer contaminant compounds than the chemical synthesis process. This also results in the production of less toxic byproducts. Biosynthesized AgNPs are biocompatible and less toxic than chemically synthesized AgNPs, which is particularly advantageous for biomedical applications. Biosynthesis generally uses precursors that are easily and affordably obtained; meanwhile, chemical synthesis is prone to requiring expensive precursors or catalysts. Biosynthesized AgNPs tend to have better stability than chemically synthesized AgNPs because the biomolecules present in the biological precursors can act as capping and stabilizing agents that prevent agglomeration and oxidation of the AgNPs. Biologically synthesized AgNPs have inherent bioactivity because they can harness the properties of the precursors; for example, if the plant extract used has already a certain antimicrobial activity, the AgNPs synthesized using this extract will have greater antimicrobial activity. Further, the use of biological synthesis agents often results in better control of the size and shape of the AgNPs, which is crucial in many applications since many properties depend on the size and shape [21,22,23].

All the above make biosynthesized silver nanoparticles an interesting alternative in many fields, particularly in biomedical and other human-related applications. It is also important to note that the characteristics of biosynthesized AgNPs are going to differ depending on the synthesis agent and method [23].

The trend observed in research is to employ processes that use a minimum amount of toxic substances, which are cheaper and have a low environmental impact. This can be achieved by using different extracts obtained from plants and their components [20,24].

*J. mimosifolia*, commonly known as the “Jacaranda” tree, is a tree native to South America that can grow up to 20 m tall, has abundant purple flowers, and is cultivated in tropical and subtropical regions due to its attractive appearance and tolerance of a wide range of growing conditions. In addition to its ornamental value, the tree has a traditional medicinal use, and as the flowers of the Jacaranda tree have a high content of polyphenolic compounds, according to Aguirre-Becerra et al. [25], therefore these compounds can act as reducers in AgNPs biological synthesis, or biosynthesis.

The aim of this work was to biosynthesize and characterize silver nanoparticles using an aqueous extract of *J. mimosifolia* flowers (JAgNPs) and compare their in vitro antibiofilm activity against chemically synthesized silver nanoparticles (QAgNPs). The in vivo antibiofilm activity was evaluated using fresh lettuce, and their antibacterial activity was tested against Gram-negative *P. aeruginosa* and Gram-positive *S. aureus* bacteria.

## 2. Results and Discussion

### 2.1. AgNPs Synthesis and Optimization

#### 2.1.1. Synthesis

The reagents and precursor used for the biological synthesis of AgNPs included an aqueous extract of *J. mimosifolia* and silver nitrate (AgNO_3_), while sodium citrate (Na_3_C_6_H_5_O_7_) was employed for chemical synthesis (Figure 1A). The color of the solution intensified after the biological synthesis reaction of the JAgNPs, developing a characteristic dark-reddish-brown color (Figure 1B) similar to the one reported by Padilla-Camberos et al. [26]. Meanwhile, QAgNPs were electrostatically stabilized using sodium citrate, which acted as a reducing and stabilizing agent, causing a color change from clear to yellowish grey (Figure 1B) [27,28]. Based on the observation that the Jacaranda aqueous extract was able to biosynthesize AgNPs, we further optimized the synthesis of both types of nanoparticles.

#### 2.1.2. Optimization of the AgNPs Synthesis

To optimize the synthesis of biological and chemical nanoparticles, a Box–Behnken design was used to evaluate the effects of independent variables on the synthesis of each reaction. For the biological synthesis of JAgNPs, pH values of 6–12, concentrations of 1–10 mM AgNO_3_, and temperatures in the range of 60–90 °C were tested. For the chemical synthesis of QAgNPs, concentrations of 1–5 mM AgNO_3_, reaction times of 30–120 min, and temperatures of 80–100 °C were evaluated. The dependent variable, or response variable, for this design was the absorbance at a wavelength of 420 nm. Table 1 shows the designs of different experiments with the independent variables and the dependent variable obtained. The absorbances presented are considering the dilution factor of samples.

Although the experimental data showed that the reactions with the highest absorbance were reaction number 14 for the JAgNPs, with an absorbance of 3.249 at 420 nm, and reaction number 11 for QAgNPs, with an absorbance of 3.864 at 420 nm, the response surface analysis revealed that the pH, silver nitrate concentration, temperature, and reaction time affected the absorbance.

For biosynthesis, the pH was found to have the greatest effect on the response variable. According to the obtained response surface (Figure 2), the optimal conditions for the synthesis of JAgNPs were pH 12, 90 °C, and 10 mM silver nitrate. For both images, it can be observed that the highest absorbance could not go from 4 (yellow) to 4.5 (orange).

**Figure 2 antibiotics-12-01084-f002:**
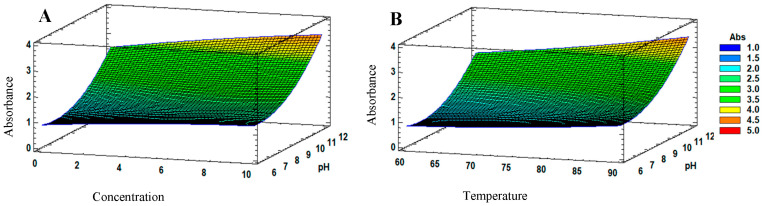
Estimated response surface of the JAgNPs: (**A**) At a predefined temperature of 90 °C, the interaction between the silver nitrate and the pH of the extract can be observed, from which the highest absorbance was provided for 10 mM silver nitrate. The changes in the absorbances are represented by a different color, from 1 (blue) to 5 (red). For chemical synthesis of QAgNPs, the response surface analysis shows that changes in silver nitrate concentration, reaction time, and temperature affect the absorbance (Figure 3), where silver nitrate concentration has the greatest effect on the response variable, followed by reaction time. The optimal conditions for the synthesis of QAgNPs were determined to be 1 mM, 100 °C, and a reaction time of 30 min. (**A**) shows the maximum absorbance value (5), represented by red, for the temperature and concentration factors, while in (**B**), the highest absorbance was orange (4.5) for the temperature and reaction time factors, with which the concentration of silver nitrate and the temperature increase the absorbance more than the reaction time of any combination.

### 2.2. Characterization

Nanoparticles have unique physical and chemical properties. Characterizing nanoparticles is essential to understand their size, shape, composition, surface properties, and aggregation behavior. This information is critical to determining their potential applications in different fields. The lack of proper characterization is challenged to control the properties of nanoparticles and adjust them to specific applications. Additionally, the characterization of nanoparticles is essential for evaluating their potential environmental and health impacts. Therefore, the characterization of nanoparticles is crucial for their safe and efficient use in various applications. Overall, the UV–Vis spectrophotometry, FTIR spectroscopy, DLS, Z potential, and TEM analyses provide valuable information about the synthesis and characterization of the JAgNPs and QAgNPs.

#### 2.2.1. UV–Vis Spectrophotometry

UV–Vis spectrophotometry was used to confirm the presence of silver nanoparticles in the samples. The spectrophotometric analysis conducted 24 h after the synthesis revealed an absorbance peak at 420 nm for the JAgNPs and at 450 nm for the QAgNPs (Figure 4), which are consistent with previously reported values in the synthesis of silver nanoparticles [26,29]. In contrast, the sodium citrate, silver nitrate, and Jacaranda flower extract did not show absorbance peaks in the range of silver nanoparticles. The absorbances considered the dilution factor.

#### 2.2.2. Fourier-Transform Infrared Spectroscopy

FTIR spectroscopy helps to identify functional groups present in the flower extract and the silver nanoparticles (Figure 5). The observed peaks are interpreted as the presence of compounds that may be functional groups from 4000 to 2000 cm^−1^ and related to fingerprints of specific compounds ranging from 1900 to 500 cm^−1^ of the infrared spectrum. FTIR spectra obtained for *J. mimosifolia* extract and JAgNPs (Figure 5) showed a peak at 3600 cm^−1^, which corresponds to an O-H phenol stretching; another one at 2900 cm^−1^ that corresponds to a C-H aldehyde stretching; a peak at 1650 cm^−1^ that is characteristic of flavonoids and aromatic rings; and a peak at 1000 cm^−1^ corresponding to carbohydrates [30]. Different types of carbohydrates may act as reducing agents in the biosynthesis of nanoparticles, as these peaks were only present in the extract. The peak at 1400 cm^−1^ in both NPs corresponded to the silver nitrate, already reported [31].

QAgNPs showed a peak in the region from 3600 to 3160 cm^−1^; this might correspond to water in the sample. There is also a peak at 2820 cm^−1^ that might be related to some C-H bonds present in the aldehyde or ketone functional groups. The identity peaks for the sodium citrate are at 1590 cm^−1^ for the asymmetric stretch of the -COO and another one at 1400 cm^−1^ for the symmetric stretch of the -COO. There is another peak at 1370 cm^−1^ that might belong to silver [30,31]. Further, at 1370 cm^−1^, there is a peak that could belong to silver or a stretch of the =C-H group [32].

Interestingly, the resulting spectra share similar characteristics of functional groups, as they have all been resuspended in water. However, differences can also be observed between the Jacaranda extract and JAgNPs with the loss and addition of compounds related to biosynthesis. There were some similarities between the JAgNPs, QAgNPs, and Jacaranda flower aqueous extract in the peaks in the region from 2900 to 2820 cm^−1^ (aldehydes and ketones), the peak at 3600 cm^−1^ (O-H), and the peak at 1645 cm^−1^ (H-O-H). These last two could be related to the presence of water in the samples, where JAgNPs have the least water quantity. The similarities and differences suggest that the components of the flower extract were involved in the reduction and stabilization of the silver nanoparticles, potentially through the functional groups identified by FTIR spectroscopy.

#### 2.2.3. Dynamic Light Scattering

DLS was used to measure the particle size of the nanoparticles, revealing that the value in the JAgNPs corresponded to 33.02 nm (Figure 6A). The particle sized is useful for assessing the stability and potential for aggregation of the nanoparticle suspensions. Previous studies have reported that most phytosynthesized nanoparticles fall within the 5–100 nm size range [33], indicating the JAgNPs are within a typical size range. However, it is worth noting that particles with different sizes were present; this can be attributed to remaining components from the herbal extract that did not react.

In contrast, the mean particle size for the QAgNPs was found to be 74.76 nm (Figure 6B), considerably larger than JAgNPs.

The polydispersity index provides information on the size distribution in mono- (<1) and polydisperse (≥1) criteria. In the case of AgNPs, values of 0.54 and 0.31 were obtained for JAgNPs and QAgNPs, respectively, which mono-disperse particles are considered in both cases. These changes in polydispersity can be related to the synthesis conditions, as has already been reported by other authors [34,35,36].

The Z potential is a measurement of the effective electric charge on nanoparticles’ surface; it is a parameter that helps to evaluate the stability of a nanoparticle suspension. If the Z potential is <−20 or >20 mV, the suspension can be considered stable, and if the Z potential is >−20 or <20 mV, the suspension is considered unstable [37]. Both JAgNPs (6C) and QAgNPs (6D) indicated stable suspensions [38]. It can also be appreciated that the QAgNPs were possibly agglomerated, as Vinothini and Rajan [37] report that NPs solution is agglomerated when the Z potential is lower than −28 mV, and this can result in the loss of biological activity.

DLS analysis is also a very important measure for biomedical applications since this will give important information on how the NPs will interact with cells, biomolecules, tissues, and other parts of biological systems. Z potential will dictate the electronegative interactions and the stability of the particles, which are crucial for drug delivery, antibacterial activity, and protein adsorption, among others. A negatively charged surface is desirable for antibiofilm and antimicrobial applications because this will allow the AgNPs to interact with positively charged bacterial membranes and other positively charged components of the extracellular matrix [38,39,40,41,42].

#### 2.2.4. X-ray Diffraction

X-ray diffraction is a technique to mainly determine the crystal structure. However, it is also used to determine the atomic arrangement, the lattice parameters, and the size; according to the calculation, the size of the QAgNPs crystal was 16.8 nm. Figure 7A shows the peak distribution pattern for the QAgNPs, which present the characteristic peaks of the silver NPs, which, according to the diffraction patterns at 38.2°, 46.3°, 34.74°, 77.23°, and 81.82° correspond to the planes (111), (200), (220), (311), and (222), respectively [43]. On the other hand, Figure 7B shows the distribution pattern corresponding to structures with patterns similar to Ag_2_O_3_, AgO described for amorphous structures [44]; under this analysis, the QAgNPs can be considered structures with good organization and crystallinity compared to the structures of JAgNPs, which can be seen by TEM micrographs.

#### 2.2.5. Transmission Electron Microscopy

The micrographs obtained by TEM showed that JAgNPs have a uniform spherical shape (Figure 8A), which is in line with the most common shape observed for phytosynthesized nanoparticles, according to Noah [33]. Even though most of the JAgNPs seem to have approximately the same size, it is desirable to have a monodispersed sample. To achieve this, purification techniques, such as density gradient centrifugation or chromatography, can be employed [45]. TEM micrographs of QAgNPs (Figure 8B) revealed irregular shapes and sizes in comparison to the JAgNPs. This characteristic has been previously reported by other authors [46] for chemically synthetized AgNPs using sodium citrate.

### 2.3. Antibacterial Activity

The Clinical and Laboratory Standards Institute (CLSI) recommends well-established methods for evaluating the effectiveness of antimicrobial agents [47]. Among these, the disk diffusion test provides a qualitative measure of the ability of nanoparticles to inhibit bacterial growth. Meanwhile, the Minimum inhibitory concentration (MIC) and Minimum bactericidal concentration (MBC) are used to determine the lowest concentration required to inhibit bacterial growth and kill bacteria, respectively. By using these three tests together, a comprehensive evaluation of the antibacterial efficacy of silver nanoparticles against various bacterial species can be achieved.

#### 2.3.1. Disk Diffusion Test

Table 2 presents the inhibition diameter values in mm of the disk diffusion test for JAgNPs and QAgNPs at different concentrations: 342 and 171 µg/mL for the JAgNPs and 235 and 117.5 µg/mL for the QAgNPs (Figure 9). In general, both types of NPs exhibited antibacterial activity, without a significant difference. The inhibition diameters for JAgNPs were found to be larger than those for QAgNPs, indicating better antibacterial activity for the former. However, the ANOVA analysis revealed that there are no significant differences between biosynthesized and chemical nanoparticles. The only differences are between the positive and negative controls for *S. aureus* and *P. aeruginosa*.

The inhibition values observed in our study were smaller than those reported by Barabadi et al. [48], who obtained higher inhibition values of 17 mm for their biosynthesized AgNPs from an aqueous *Zataria multiflora* extract and 14 mm for commercially available AgNPs. In addition, the observed difference in inhibition diameter values for *S. aureus* between our study and that mentioned above could be attributed to the higher microbial suspension concentration used in our work (10^9^ vs. 10^6^ CFU/mL). This difference indicates that our study used a higher bacterial suspension, potentially leading to smaller inhibition diameters, and the difference in the ATCC strain could have played a role (6538 and 25,923, respectively).

Some research has evaluated the differences between the inhibition values of biosynthesized and chemically synthesized AgNPs. One example is the study conducted by John et al. [49], where they tested the microbial inhibition activity of biologically synthesized AgNPs vs. chemically synthesized AgNPs in Gram-positive and Gram-negative bacteria. They proved that biosynthesized AgNPs have higher biological activity than those chemically synthesized.

Although our JAgNPs exhibited antibacterial activity, it is worth considering the potential impact of the synthesis time on their efficacy. Although this was not considered in the present study, Mohan et al. [50] observed that the inhibition diameter of nanoparticles can increase from 8 to 12 mm over a synthesis period ranging from 1 to 48 h. Therefore, increasing the synthesis time might result in larger inhibition diameters.

#### 2.3.2. Minimum Inhibitory Concentration and Minimum Bactericidal Concentration

The JAgNPs exhibited greater antibacterial activity, as the MIC and MBC values were lower than the QAgNPs’ (Table 3), likely due to their smaller particle size (Figure 6) and subsequent cell wall penetration. This is consistent with the findings of other studies, where a relationship has been observed between the size of AgNPs and the MIC obtained. For example, for AgNPs of chemical synthesis with a size of 20 nm, a MIC of 2.28 μg/mL was observed for *P. aeruginosa*, while another study with AgNPs of biological synthesis and a size of 5 nm showed a MIC of 0.59 μg/mL for *P. aeruginosa*, while for *S. aureus*, it showed a MIC value of 0.75 [51,52].

For this study, JAgNPs had an average particle size of 33.02 nm, while the QAgNPs had an average particle size of 74.76 nm. The reduced size of JAgNPs may enhance their ability to penetrate bacterial cells, leading to intracellular damage and subsequent cell death. This could explain the observed difference in effectiveness compared to the QAgNPs, highlighting the role of size rather than the nature of the treatments. Additionally, the detrimental effect of silver ions present on the nanoparticle surface further contributes to their antimicrobial activity. A similar mechanism was proposed by Loo et al. [53], suggesting that the size-associated effectiveness is attributed to the ease of smaller NPs passing through the cell wall and causing irreversible damage. Thuc et al. [54] evaluated the effectiveness of AgNPs at concentrations ranging from 0 to 100 µg/mL, and with microbial suspensions ranging from 10^1^ to 10^10^ CFU/mL of *S. aureus* (ATCC 25922) and *P. aeruginosa* (ATCC 27853), finding that the higher microbial concentrations required higher AgNPs concentrations to be effective.

In the present investigation, the concentrations of the microbial suspension used were higher than those used in other studies, such as that of Arsène et al. [55], where for their NPs biosynthesized with grapefruit peel, they used *S. aureus* ATCC 6538, the same strain used in this study, but with a concentration lower than 10^8^ CFU/mL, finding a MIC of 6.5 µg/mL. In contrast, this study found a MIC of 5.3 µg/mL for the same strain, but with a higher concentration of 10^9^ CFU/mL. The NPs reported by Arsène et al. [55] had a larger particle size (160.1 nm), which supports the size-related efficacy. It could mean that the JAgNPs tested in less concentrated microbial suspensions could be more effective.

It is also important to note that the particle size is not the only factor to consider when evaluating and comparing the effectiveness of different NPs, as Mohan et al. [50] tested different time intervals (1, 5, 18, 24, and 48 h) and found that with 1 h of synthesis, the MIC was 10 µg/mL, and with 48 h, the MIC decreased to 6 µg/mL. Further experiments could explore this interesting factor.

#### 2.3.3. Biofilm Determination by the Congo Red Method

The Congo red method was used to qualitatively determine biofilm formation. Dark-brown or black crystals were interpreted as biofilm formation, while pink or light-brown colonies indicated the inhibition of biofilm formation. Figure 10A–C show the differences in biofilm formation between the untreated *S. aureus* ATCC 6538 control and the same species treated with the MIC of JAgNPs and with QAgNPs, respectively.

Figure 10D–F illustrate the differences in biofilm formation between the untreated *P. aeruginosa* ATCC 9027 control and the same bacteria treated using the MIC of JAgNPs and QAgNPs, respectively. In Figure 10, we observed that the control sample of *S. aureus* shows a growth of blackish colonies compared to the red growth in the nanoparticles. The same effect was observed.

After treatment with both AgNPs, the absence of biofilm was observed in both bacterial species. Furthermore, when the bacterial species were treated with the MBC, there was a complete absence of growth in all cases, which is consistent with the findings of other studies [56,57,58,59,60].

The presence of biofilm is associated with the expression of *Ica ABCD* genes, which are included in a pool of tests used for biofilm detection, although some studies by Ghostaslou et al. [59] and Leshem et al. [58] have shown variations in the sensitivity of tests, such as the Congo red test and the microplate test. Additional complementary tests that incorporate *Ica ABCD* gene expression might be necessary. Nevertheless, the results obtained from this study demonstrate that both types of AgNPs were highly effective at preventing biofilm formation in tested bacterial species.

#### 2.3.4. Antibiofilm Activity by Crystal Violet Method

The microplate biofilm inhibition results for both *S. aureus* and *P. aeruginosa* bacterial species are shown in Figure 11. JAgNPs exhibited higher inhibition values in comparison with the QAgNPs. For both types of AgNPs, three different concentrations were tested. JAgNPs were tested at concentrations of ½ MIC, MIC, and 2 MIC (2.67, 5.35, and 10.7 µg/mL) and yielded inhibition rates of 81.9, 95.9, and 96.3% against *S. aureus* and 26.8, 96.3, and 96.9% against *P. aeruginosa*. In contrast, QAgNPs were tested at the concentrations ½ MIC, MIC, and 2 MIC (58.75, 117.5, and 235.0 µg/mL) and yielded inhibition rates of 13.6, 9.67, 0.37, 9.67, and 13.16% against *S. aureus* and 63.56, 75.75, and 76.14% against *P. aeruginosa*.

The statistical analysis revealed that for *S. aureus*, there is a significant difference be-tween JAgNPs and QAgNPs at all concentrations tested; however, JAgNPs did not show a significant difference among their MICs tested, and the same case for QAgNPs.

With respect to *P. aeruginosa*, JAgNPs present a significant difference between their MICs with QAgNPs; however, for JAgNPs, there is only a significant difference between 1/2 MIC with the other 2 concentrations of JAgNPs tested, while for QAgNPs, there is no significant difference between their MICs. In *P. aeruginosa*, outlier standard deviations, such as within 1/2 MIC of JAgNPs for *P. aeruginosa*, may be due to the number of repetitions in the test.

In this study, the JAgNPs biosynthesized demonstrated higher inhibition values than reported in a previous study [61], despite using similar concentrations (2, 5, 10, and 15 µg/mL), as in this study (2.67, 5.35, and 10.7 µg/mL); achieved inhibition percentages of 18.8, 36.1, 62.0, and 77.6%, with AgNPs biosynthesized with *Carum copticum* with an average size of 21.48 nm, while in this study, the percentages were 26.8, 96.3, and 96.9% for *P. aeruginosa*. Other studies have reported a limited effectiveness of NPs against the formation of biofilms in *P. aeruginosa*, with inhibition percentages that barely reach 75%, such as that of Lewis et al. [62], where they obtained a 67% inhibition using 100 µg/mL of AgNPs biosynthesized with *Spirulina platensis* extract with a size of 29 nm, or by Mariadoss et al. [63], who obtained 72% inhibition using a concentration of 500 µg/mL with AgNPs of biological synthesis with *Malus domestica* and a particle size of 50–107 nm.

The tendency of antibiofilm activity of JAgNPs displayed an asymptotic line. This concurs with previous observations by Singh et al. [64] when testing different *Cannabis sativa*-biosynthesized AgNPs with a size of 20–40 nm against biofilm forming *E. coli* and *P. aeruginosa*.

In the case of *S. aureus*, the biofilm inhibition values obtained are consistent with previous reports in the literature. For example, Bharati et al. [65] observed an antibiofilm activity of 92% using a concentration of 100 µg/mL for AgNPs biosynthesized with extract of *Cordia dichotoma fruits* and a size of 2–60 nm.

#### 2.3.5. Antibiofilm Activity in Lettuce Leaves

The results for the in vivo antibiofilm activity of AgNPs are presented in Figure 12. JAgNPs had higher logarithmic (Log) reduction values than QAgNPs for *S. aureus* and *P. aeruginosa* bacterial species when the MIC of both AgNPs in both strains was analyzed (MIC: 5.3 and 117.5 µg/mL). The biosynthesized JAgNPs showed a log reduction of 0.63 and 2.38 for *S. aureus* and *P. aeruginosa,* respectively, while the chemically synthesized QAgNPs presented a log reduction of 0.006 and 0.41 for the same bacterial species.

To assess the effectiveness of the logarithmic CFU/mL reduction, it is important to consider the bacterial species used in the study. For example, Hussaein et al. [66] used *Bacillus cereus* and found that a 4 Log reduction is optimal. Srey et al. [67] used sodium hypochlorite as a positive control and defined that a 3 Log reduction was considered effective. In this study, we obtained a 2.38 Log reduction, a near value when using JAgNPs against *P. aeruginosa*.

In terms of the mechanism for the antibiofilm activity, JAgNPs may interfere with the initial attachment of bacteria to surfaces, disrupting the formation of biofilm matrices. The nanoparticles might bind to the bacterial cell wall, preventing the adhesion of bacteria to surfaces and inhibiting biofilm formation. Moreover, JAgNPs could enter the bacterial cells and disrupt the intracellular processes necessary for biofilm development and maintenance [7].

As mentioned, the factors influencing the results can be related not only to the particle size, but also to other factors, such as the ATCC, inoculum concentration, shape of the NPs, or residual compounds from the botanical extract. The known mechanisms responsible for the bactericidal effect of NPs include the ability to bind to the cell wall to limit cellular respiration, to inhibit the binding of other compounds, to enter the cell wall and induce cell death, or to release silver ions into the cytoplasm, interfering with cellular processes resulting in a bacteriostatic effect [53].

The results obtained in this study suggest the need for additional testing with a variety of biofilm-forming species, such as *E. coli* O157H7, *Salmonella enterica*, methicillin-resistant *S. aureus* (MRSA), and *Bacillus cereus* [1], using a similar inoculum concentration reported in other studies, homogenous results showing strong evidence concerning the antibacterial and antibiofilm activity of the tested AgNPs under different circumstances. This will enhance our comprehension of the underlying mechanisms responsible for the bactericidal and antibiofilm effects of JAgNPs and their potential use as an alternative to conventional chemical disinfectants in food industries.

## 3. Materials and Methods

### 3.1. Synthesis and Optimization of AgNPs

#### 3.1.1. Chemical and Biological Synthesis

The chemical and biological syntheses were carried out using the reducing agents sodium citrate and *J. mimosifolia* flower aqueous extract, respectively. Changes in color from transparent to yellowish grey and dark reddish brown for the chemical and biological syntheses, respectively, were indicators of AgNPs formation [26,27,28].

For the aqueous extract of *J. mimosifolia*, the pulverized plant was mixed with distilled water in a 1/100 ratio. Briefly, 100 mL of distilled water was brought to boil, and 1 g of the plant was added. This was maintained at 92–100 °C for 5 min. The remains of any organic matter were removed with a filter paper of 11 µm, and the resulting extract was stored at 4 °C until use. For the chemical synthesis, a previously studied concentration of sodium citrate (0.04 g/L) was added [68] to the different conditions evaluated, which included varying temperature, pH, and silver nitrate concentration.

Due to the influence of different factors in the formation of nanoparticles, an experimental design was carried out to establish the optimal conditions of the synthesis process [68].

#### 3.1.2. Optimization of the Synthesis of AgNPs

To identify how the interaction among the factors influenced the synthesis reaction, a Box–Behnken design was used in Statgraphics software version XV, where the independent variables to be evaluated were temperature, silver nitrate concentration, pH (only the biological synthesis), and reaction time (only for chemical synthesis). Thus, 15 combinations were made with 3 repetitions in the central point and 3 levels of each independent variable, with values of 60, 75, and 90 °C (biological synthesis); 80, 90, and 100 °C (chemical synthesis); 1, 5.5, and 10 mM (biological synthesis); 1, 3, and 5 mM (chemical synthesis); pH 6, 9, and 12 (biological synthesis); and 30, 75, and 120 min (chemical synthesis). We defined the dependent variable, or response variable, as the highest absorbance at a wavelength of 420 nm. The optimal conditions obtained from the Box–Behnken method were evaluated [69].

### 3.2. Characterization of the AgNPs

#### 3.2.1. Spectrophotometry UV–Vis

A spectral scan was performed from 300 to 700 nm with 10 nm intervals, using a Bio Rad xMark Microplate spectrophotometer to measure the absorbance peak and confirm the presence of AgNPs [26].

#### 3.2.2. Fourier-Transform Infrared Spectroscopy

The functional groups present in the *J. mimosifolia* extract and in the AgNPs were evaluated by FTIR. The samples were lyophilized; subsequently, enough sample was placed to cover the glass of the ATR accessory of the Cary 630 Agilent FTIR, and the readings were performed with a resolution of 4 cm^−1^ and 20 scans [30].

#### 3.2.3. Dynamic Light Scattering

The size and stability of the nanoparticles were determined with a Malvern Nano-ZS90 Zetasizer. The AgNPs obtained by chemical and biological syntheses were transferred to the plastic cells. The data on the material to be analyzed and the dispersant were loaded into the program before taking the readings. Between each measurement, the cells were washed 10 times with distilled water [26].

#### 3.2.4. X-ray Diffraction (XRD) Analysis

X-ray diffraction (XRD) analysis was performed using a D2 PHASER of Bruker powder diffractometer for operating with a step size of 0.02, with a time of 0.8 s per step. A range of 10 to 80 degrees in 2 theta was measured; the sample was placed in the center of an acrylic sample holder and broken during the measurement at an angular velocity of 1°/min. The average crystal size can be determined using the Debye–Scherrer equation Equation (1), with the most intense peak of the diffraction pattern found at the 37.9° position [43,44].
D = kλ/βcosθ(1)
where D indicates the average crystal size in nm; λ is 0.15418 nm, which corresponds to the CuKα tube of the diffractometer; and β is the FWHM (Full Width at Half Maximum) of the most intense peak of the diffraction pattern with instrumental width correction.

#### 3.2.5. Transmission Electron Microscopy

The nanoparticle samples were sent to the Instituto Potosino de investigación científica y tecnológica (IPICYT) and analyzed using a Jeol model 200CX microscope. The samples were sonicated for 5 min and allowed to settle for 10 min before introducing them under the microscope [26].

### 3.3. Antibacterial Activity of the AgNPs

To evaluate the antibacterial activity in all assays, the bacterial species *S. aureus* ATCC 6538 (Gram-positive) and *P. aeruginosa* ATCC 9027 (Gram-negative) were used. The data obtained were analyzed with ANOVA in Statgraphics XV, considering a significant difference a *p* value = 0.05 for all assays.

#### 3.3.1. Disk Diffusion Test

The disk diffusion method was used to calculate the zone of inhibition [70]. In this method, each species was seeded uniformly on nutrient agar plates using sterile swabs. Subsequently, a period of 20 min was allowed to elapse for the excess water in the bacterial suspension to evaporate completely. Then, dry Whatman (6 mm) filter paper discs were placed that had previously been treated (by immersion in liquid and subsequently dried) with JAgNPs at 342 and 171 µg/mL and QAgNPs at 235 and 117.5 µg/mL. The dry mass of each sample was reconstituted with a determined volume of water, then the reconstituted sample and a dilution in half were tested, so the concentration of each sample used for antimicrobial activity tests is different [26]. The paper discs were gently placed on the nutrient agar plates and incubated at 37 °C for 24 h. The diameter of inhibition zones around the discs were measured in mm, subtracting the diameter of the filter paper disk, and the result of the disc diameter (6 mm) was normalized with the value obtained from the control. Assays were performed in duplicate. The controls used were erythromycin for *S. aureus* and ciprofloxacin for *P. aeruginosa*.

#### 3.3.2. Determination of the Minimum Inhibitory Concentration (MIC) and Minimum Bactericidal Concentration (MBC)

The bacterial inhibition effect of AgNPs was evaluated with the Minimum inhibitory concentration test [26]. Thus, a standard procedure of microdilution in broth with 48-well polystyrene plates was used. Briefly, 20 µL of the culture of each standardized strain (0.5 McFarland) was mixed with 500 µL of nutrient broth containing different concentrations of JAgNPs (1.3–342 µg/mL) and QAgNPs (0.9–235 µg/mL), then the plates were incubated at 37 °C for 24 h. The MIC was determined as the concentration that visually avoided the growth of the bacteria in comparison to the control (non-treated). For the MBC, nutrient agar plates were used, in which 50 µL of the suspensions of each well of the 48-well plates used for the MIC was spread and incubated at 37 °C for 24 h. The MBC was identified as the lowest concentration that killed bacteria on the nutrient agar plates. The assays were performed in triplicate.

#### 3.3.3. Determination of Biofilm Formation by Congo Red

The qualitative determination of biofilm formation was carried out using the method of Kalishwaralal et al., with modifications [56]. For this, the standard brain–heart agar was used at 37 g/L. The medium was sterilized at 121 °C/15 min, and when the medium reached 55 °C, 0.08% *w*/*v* of Congo red, 5% *w*/*v* sucrose, chloride sodium 1.5% *w*/*v*, and glucose 2% *w*/*v* were added [57]. When the homogeneity of the medium was observed, it was divided into 3 sterile flasks. The JAgNPs were added to the first vial with medium and QAgNPs to the second until reaching the MIC in both cases, while nothing was added to the third. Media from all 3 vials were plated and allowed to solidify—after that, the *S. aureus* and *P. aeruginosa* bacterial species. The plates were incubated at 37 °C for 24 h, and the changes in the color and appearance of the colonies were evaluated. The presence of black and brown colonies was taken as positive for biofilm, indicating the production of exopolysaccharides by the bacterial species, and negative when colonies were red.

#### 3.3.4. Microplate Antibiofilm Activity Method

Antibiofilm activity was assessed using the crystal violet method [71]. In a sterile 96-well microplate, 100 µL of JAgNPs (2.7, 5.35, and 10.7 µg/mL) and QAgNPs (58.75, 117.5, and 235 μg/mL) were added and mixed with nutrient broth. Subsequently, 20 µL of the bacterial species were inoculated per well and allowed to incubate at 37 °C for 24 h. After incubation, the wells were washed with phosphate buffer saline (PBS, pH 7.2) to remove non-adhered cells. The plates were dried at 60 °C, after which 100 µL of 0.1% crystal violet was added to stain the biofilms. The dye was removed from the wells after 90 min. This was followed by 2 washes with PBS. The attached cells were fixed by adding 100 µL of 96% methanol per well. After stirring for 10 min, the absorbance was read at 570 nm with a UV–Vis a BioRad xMark Microplate spectrophotometer. The percentage of biofilm inhibition was calculated by comparison with the untreated control, which was considered as 100% of biofilm formation:% Biofilm Inhibition = (1 − (SA/CA)) × 100 
where SA stands for sample absorbance, and CA is the control absorbance. As positive control, multi-enzymatic detergent (1% *w*/*v*) and nutrient broth (growth control) were used. The assays were performed in duplicate.

#### 3.3.5. Antibiofilm Activity on Fresh Lettuce

To evaluate the antibiofilm activity present in vegetables, the methodology of Turhan et al. and Klug et al. [72,73] was used.

For the test, fresh leaves of lettuce (*Lactuca sativa*) purchased locally were used. The inner leaves close to the stem were chosen to cut the coupons (1 cm), and these were later washed and disinfected with chlorine (0.75% *v*/*v*) to reduce the microbial load. Coupons were inoculated with 50 µL of the bacterial suspension in 6-well flat-bottom plates. These were subsequently sealed and incubated at room temperature for 24 h. After the incubation period, the coupons were washed with PBS and sterile water to eliminate nonadherent bacteria and treated with JAgNPs and QAgNPs at the MIC for 10 min. Subsequently, Na_2_S_2_O_3_ was used to neutralize the AgNPs activity. The biofilm-forming residual strain was recovered in 15 mL conical tubes with 9 mL of peptone (0.1% *w*/*v*), with constant agitation for 3 min to detach and homogenize the biofilm. Serial dilutions were made in PBS and plated on nutrient agar for plate counting after an incubation period at 37 °C for 24 h. The assays were performed in duplicate, using the strain treated with PBS as a growth control. The results are expressed as the Log CFU/mL reduction. The tests were performed in duplicate.

## 4. Conclusions

This study demonstrated the feasibility of using a Jacaranda flower extract to biosynthesize AgNPs with enhanced bioactivity with an optimized process. The JAgNPs showed superior antibacterial and biofilm inhibition activity compared to chemically synthesized nanoparticles, attributed to their reduced size, which enhances penetration into bacterial cells, leading to intracellular damage and cell death. Moreover, JAgNPs disrupt the initial attachment of bacteria to surfaces, inhibit biofilm formation, and interfere with intracellular processes necessary for biofilm development and maintenance. The JAgNPs showed greater biofilm inhibition activity in both artificial materials and fresh lettuce, compared to QAgNPs, suggesting their potential use as a biofilm-preventing agent in the food industry. This opens the possibility of using JAgNPs to formulate sanitizing products that prevent the formation of biofilms; however, more studies are necessary to evaluate the stability of the AgNPs, their activity against other microorganism of interest in the food industry, time kill tests, their potential as a surface sanitizer, and their application in other areas and medical supplies. Moreover, these findings provide insight into the potential of utilizing plant extracts for the synthesis of AgNPs with enhanced bioactivity and offer a promising and effective alternative to traditional chemical disinfectants associated with health concerns.

## Figures and Tables

**Figure 1 antibiotics-12-01084-f001:**
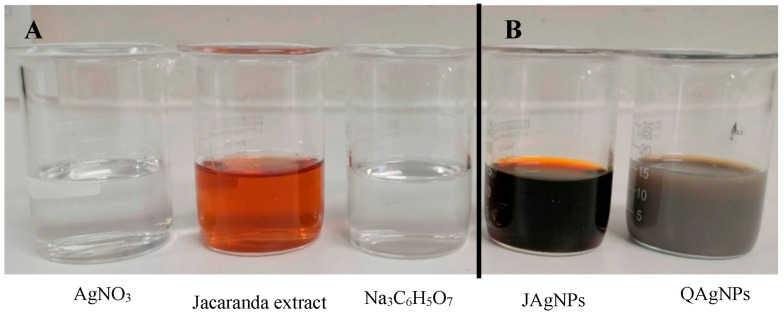
(**A**) Precursors: Jacaranda extract (*J. mimosifolia*)*,* silver nitrate (AgNO_3_), and sodium citrate (Na_3_C_6_H_5_O_7_); (**B**) silver nanoparticles biosynthesized with Jacaranda extract (JAgNPs) and chemical synthesis (QAgNPs). The changes in the colors of the precursors, from transparent and light brown to dark reddish brown for the JAgNPs and from transparent to yellowish gray for the QAgNPs, reveal the production of nanoparticles.

**Figure 3 antibiotics-12-01084-f003:**
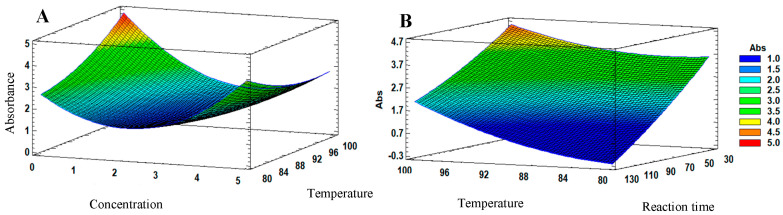
Estimated response surface of QAgNPs: (**A**) at a predefined reaction time of 75 min, the interaction between the silver nitrate and the reaction temperature can be observed, where 1 mM and 100 °C are the factors that resulted in the highest absorbance; (**B**) interaction between the reaction time and temperature concerning the increase in absorbance, with a predetermined concentration of 1 mM. The changes in the absorbances are represented by colors ranging from blue (1 abs) to red (5 abs).

**Figure 4 antibiotics-12-01084-f004:**
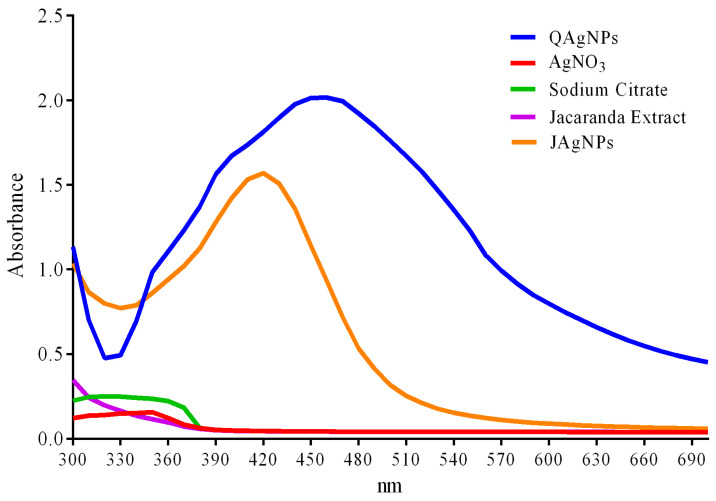
UV–Vis spectrophotometry of JAgNPs and QAgNPs, as well as the precursors *J. mimosifolia* extract and sodium citrate.

**Figure 5 antibiotics-12-01084-f005:**
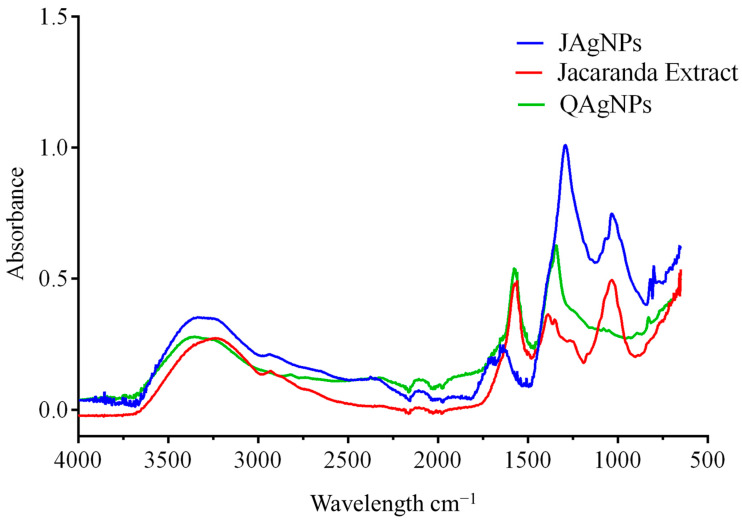
FTIR spectra of Jacaranda extract, JAgNPs, and QAgNPs.

**Figure 6 antibiotics-12-01084-f006:**
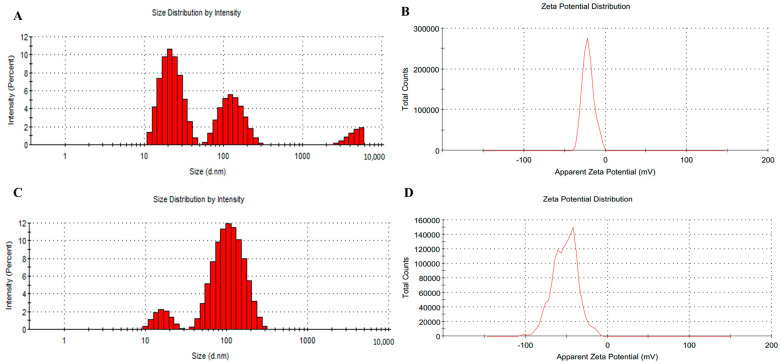
The size distribution observed, with the letters A and C for JAgNPs and QAgNPs, respectively. The Z potential is represented with the letters B and D for JAgNPs and QAgNPs, respectively. (**A**) The size distribution in percentage of intensity of the JAgNPs. Each peak represents a size in nanometers, highlighting the presence of AgNPs of various sizes, with many of the peaks at 22.64 nm, with an average size of 33.02 nm. (**B**) Z potential of JAgNPs, with a peak of −21.1 mV. (**C**) The size distribution in relation to the percentage of intensity of the QAgNPs in which only 2 peaks are observed: 1 at 16.56 nm and a second with greater intensity at 115.6 nm, with an average size of 74.76 nm. (**D**) QAgNPs with 3 peaks of −43.2, −66.3, and −102 mV and areas of 62.4, 37.1, and 0.4%, respectively, with an average Z value of −51.7 mV.

**Figure 7 antibiotics-12-01084-f007:**
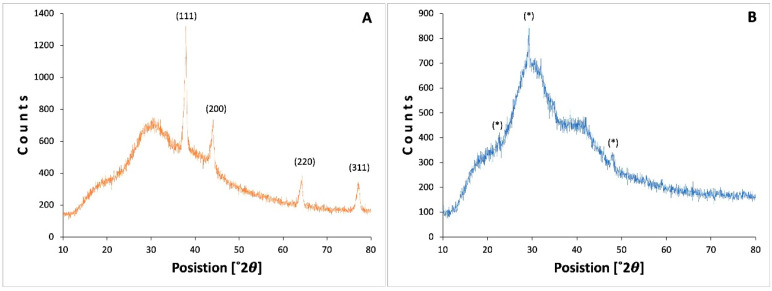
X-ray diffraction of QAgNPs (**A**) and JAgNPs (**B**) where asterisks show of peaks at 27.9°, 32.2°, and 54.72° corresponding to planes (100), (111), and (220).

**Figure 8 antibiotics-12-01084-f008:**
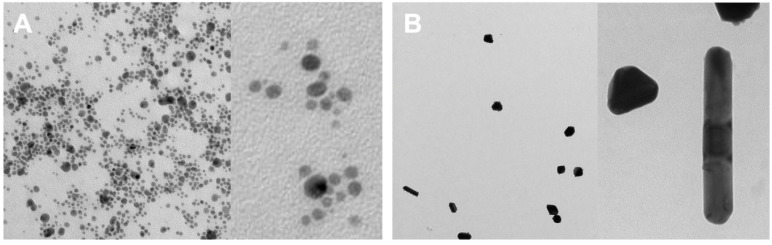
Images obtained by TEM: (**A**) JAgNPs; (**B**) QAgNPs.

**Figure 9 antibiotics-12-01084-f009:**
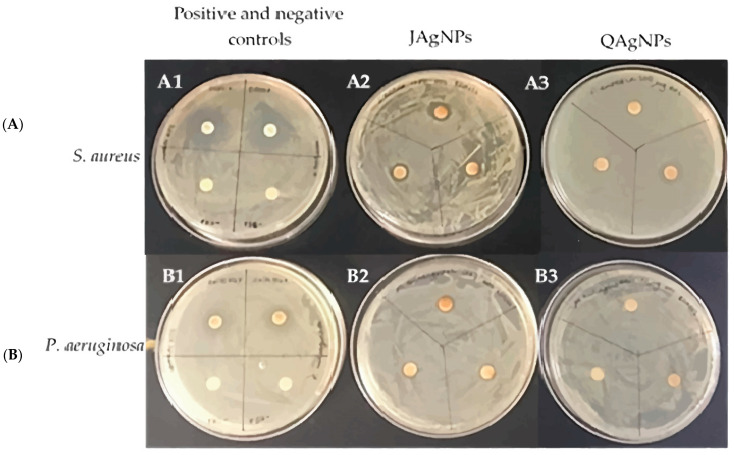
Disc diffusion results of controls (**A1**,**B1**), JAgNPs (**A2**,**B2**), and QAgNPs (**A3**,**B3**) against *S. aureus* (**A**) and *P. aeruginosa* (**B**).

**Figure 10 antibiotics-12-01084-f010:**
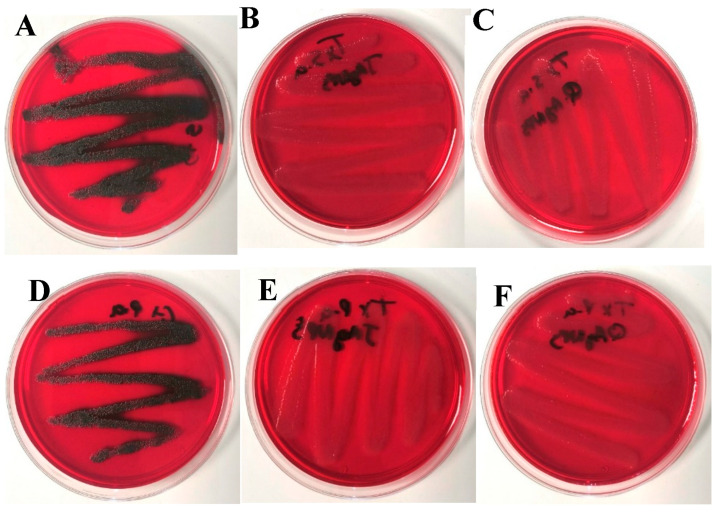
Biofilm formation in *S. aureus* (**A**) vs. MIC of JAgNPs (**B**) and QAgNPs (**C**). Biofilm formation in *P. aeruginosa* (**D**) vs. MIC of JAgNPs (**E**) and QAgNPs (**F**).

**Figure 11 antibiotics-12-01084-f011:**
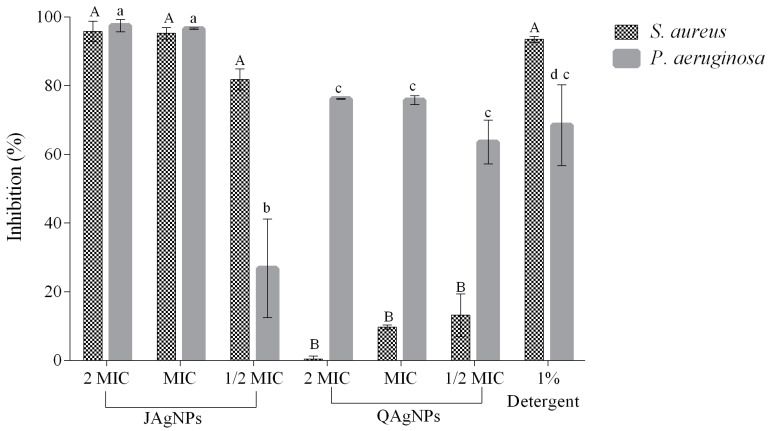
Antibiofilm activity by crystal violet method of JAgNPs and QAgNPs. The positive control was used as a multi-enzymatic detergent capable of eliminating biofilm. The concentrations for AgNPs were ½ MIC, MIC, and 2 MIC, with values for JAgNPs of 2.67, 5.35, and 10.7 μg/mL, respectively, and for QAgNPs, 58.75, 117.5, and 235 μg/mL respectively. Capital letters correspond to *S. aureus*, and lowercase letters correspond to *P. aeruginosa*. Different letters represent significant differences. Comparisons between the MICs by group (JAgNPs and QAgNPs): we can only find one significant difference between ½ MIC of JAgNPs in *P. aeruginosa* versus 2 MIC and MIC.

**Figure 12 antibiotics-12-01084-f012:**
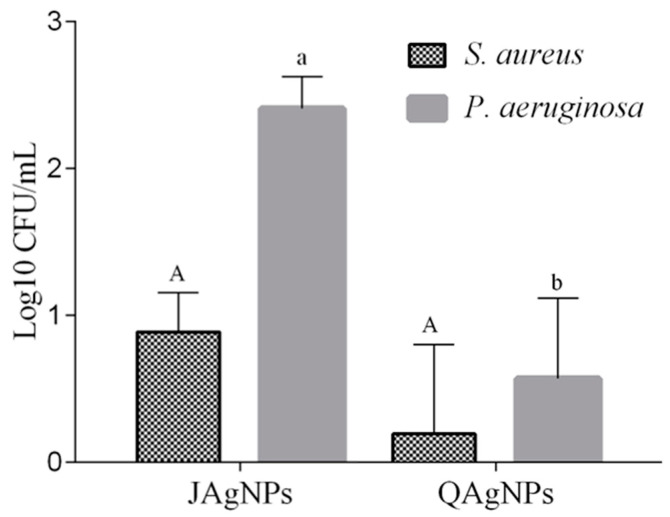
Antibiofilm activity of JAgNPs and QAgNPs in lettuce leaves for *S. aureus* and *P. aeruginosa*. The capital letters correspond to *S. aureus* and lowercase letters to *P. aeruginosa*. Different letters represent significant differences.

**Table 1 antibiotics-12-01084-t001:** Design of the experiments and absorbances recorded at 420 nm for each combination of factors.

Biological Synthesis				
Combination	pH	Concentration	Temperature	Absorbance 420 nm
	Extract	AgNO_3_ (mM)	°C	
1	6	1	75	0.772
2	9	5.5	75	1.106
3	9	1	60	0.816
4	9	5.5	75	1.127
5	12	5.5	60	2.045
6	9	1	90	1.064
7	9	10	90	1.568
8	9	10	60	1.051
9	9	5.5	75	1.106
10	12	1	75	2.111
11	6	10	75	1.054
12	6	5.5	90	1.288
13	12	10	75	2.863
14	12	5.5	90	3.249
15	6	5.5	60	0.719
**Chemical Synthesis**				
**Combination**	**Concentration**	**Time Reaction**	**Temperature**	**Absorbance** **420 nm**
	**AgNO_3_ (mM)**	**Min**	**°C**	
1	3	75	90	1.243
2	1	120	90	0.673
3	3	75	90	1.252
4	3	75	90	1.344
5	1	30	90	3.413
6	3	30	80	2.555
7	1	75	100	3.303
8	3	30	100	2466
9	3	120	80	0.654
10	5	120	90	3.693
11	5	75	80	3.864
12	3	120	100	1.756
13	5	75	100	2.507
14	1	75	80	1.982
15	5	30	90	2.826

Results of the design of the experiments for the JAgNPs and QAgNPs. The response variables for JAgNPs included the pH of the extract (6–12), silver nitrate (1–10 mM), and temperature (60–90 °C). For QAgNPs, the following variables were used: concentrations of silver nitrate (1–5 mM), reaction time (30–120 min), and temperature (80–100 °C). The absorbance at 420 nm in both cases was considered a dependent variable. The higher the absorbance, the greater the presence of AgNPs.

**Table 2 antibiotics-12-01084-t002:** Antibacterial activity of JAgNPs and QAgNPs (in µg/mL) by disk diffusion (in mm).

Bacteria	Control+	Control−	JAgNPs		QAgNPs	
			342	171	235	117.5
** *S. aureus* **	8.35 ± 0.07 ^A^	0.05 ± 0.007 ^B^	2.1 ± 0.1 ^C^	1.5 ± 1.32 ^C^	0.2 ± 0.1 ^C^	0.2 ± 0.1 ^C^
** *P. aeruginosa* **	5.2 ± 0.14 ^a^	0.05 ± 0.0007 ^c^	1.6 ± 0.51 ^c^	1.13 ± 0.32 ^c^	0.2 ± 0.1 ^c^	0.2 ± 0.1 ^c^

Results are presented as mean and standard deviation. As positive controls for the inhibition halos, the antibiotic erythromycin was used for *S. aureus* and ciprofloxacin for *P. aeruginosa*. As negative control, distilled water was used. Capital letters correspond to *S. aureus*, and lowercase letters correspond to *P. aeruginosa*. Different letters represent significant differences. The only statistic differences (*p* = 0.05) is observed between positive and negative controls for both bacterial species.

**Table 3 antibiotics-12-01084-t003:** Antibacterial activity of the JAgNPs and QAgNPs (in µg/mL) by Minimum inhibitory concentration (MIC) and Minimum bactericidal concentration (MBC).

Bacteria	JAgNPs		QAgNPs	
	MIC	MBC	MIC	MBC
*S. aureus*	5.3	10.7	117.5	235
*P. aeruginosa*	5.3	10.7	117.5	235

The tests were carried out in triplicate.

## Data Availability

Data will be made available upon request.

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
