# Peer review of "Antibacterial and Antibiofilm Activity of Chemically and Biologically Synthesized Silver Nanoparticles"

_antibiotics, 2023, doi:10.3390/antibiotics12071084_

Round 1

Reviewer 1 Report

1. Authors need to perform XRD to confirm the formation of silver nanoparticles. Also, mention the ICCD database number of AgNPs.

2. Authors need to calculate the crystal size of AgNPs based on the XRD data to understand how the synthesis method affects the size of particles.

3. Authors need to discuss and elaborate more on how biosynthesis and chemical synthesis methods of AgNP affect their Z potential value. Explain the importance of Zeta potential in biomedical applications.

https://doi.org/10.1016/j.inoche.2020.108228

https://doi.org/10.1080/10667857.2021.1965701

4. The introduction is good but very general. Authors need to provide more insight into the importance of green synthesis for biomedical applications like easy-to-control shape and size of particles, how the shape and size of particles affect biomedical applications, increase biocompatibility, bioavailability, and non-toxic byproducts, Refer https://doi.org/10.3390/antibiotics12030564 

https://doi.org/10.1016/j.bbrc.2022.04.003

https://doi.org/10.3390/biomedicines10112792

https://doi.org/10.1049%2Fnbt2.12078

5. The original images of the disc diffusion assay need to be added to the main manuscript for the reference of readers' interest. 

6. The conclusion has to be improved with key findings with updated IC50 value, mechanism beyond the activity etc.

7. Typographic errors need to be corrected. The language and grammar used throughout the manuscript need to be improved.

8. An overall graphical representation may be added in the interest of readers. 

Minor spelling and syntax errors need to addressed. overall the need some improvement in the representation.

Author Response

  1. Authors need to perform XRD to confirm the formation of silver nanoparticles. Also, mention the ICCD database number of AgNPs.

Response: We carry out an X-ray diffraction analysis through an external laboratory

  1. Authors need to calculate the crystal size of AgNPs based on the XRD data to understand how the synthesis method affects the size of particles.

Response: The crystal size of the nanoparticles was determined with the XRD

  1. Authors need to discuss and elaborate more on how biosynthesis and chemical synthesis methods of AgNP affect their Z potential value. Explain the importance of Zeta potential in biomedical applications.

https://doi.org/10.1016/j.inoche.2020.108228

https://doi.org/10.1080/10667857.2021.1965701

Response: We explain applications of Zeta potential and add references suggested

  1. The introduction is good but very general. Authors need to provide more insight into the importance of green synthesis for biomedical applications like easy-to-control shape and size of particles, how the shape and size of particles affect biomedical applications, increase biocompatibility, bioavailability, and non-toxic byproducts, Refer https://doi.org/10.3390/antibiotics12030564

https://doi.org/10.1016/j.bbrc.2022.04.003

https://doi.org/10.3390/biomedicines10112792

https://doi.org/10.1049%2Fnbt2.12078

Response: We rewrite the introduction and add references suggested

  1. The original images of the disc diffusion assay need to be added to the main manuscript for the reference of readers' interest.

Response: Imagen was included

  1. The conclusion has to be improved with key findings with updated IC50 value, mechanism beyond the activity etc.

Response: the conclusions were improved

  1. Typographic errors need to be corrected. The language and grammar used throughout the manuscript need to be improved.

Response: Typographic errors were corrected. We have made an English editing service from MDPI

  1. An overall graphical representation may be added in the interest of readers.

Response: The graphical abstract was included

Reviewer 2 Report

In the study “Antibiofilm Activity of Chemically and Biologically Synthesized Silver Nanoparticles”, the authors investigate the activity of plant extract to fabricate Ag-NPs compared to chemically synthesized ones. The synthesized AgNPs (biological or chemical) were characterized by UV-Vis, FTIR, XRD, TEM, DLS, and Zeta potential. The optimizing factor for AgNPs synthesis was studied using Box–Behnken design.  The antimicrobial activity and antibiofilm using S. aureus and P. aeruginosa were also investigated. The manuscript introduces important data but needs major revision before being accepted in the antibiotic journal.   

1-    The title should be improved to indicate the work, authors investigate antimicrobial and antibiofilm.

2-    The abstract should be rephrased to contain promising data for characterizations.

3-    “gram-negative” and “gram-positive” should be “Gram-negative” and “Gram-positive”, please revise throughout the manuscript.

4- The introduction should be improved by referring to the importance of biosynthesized AgNPs using different biological entities followed by those synthesized by plant extract. I recommend to citing the following references: https://doi.org/10.3390/catal12050462; https://doi.org/10.3390/jof8040396.

5-    Line 531, the scientific names must be in italics, please check and revised throughout the manuscript.

6-    In line 526, the authors investigate the antimicrobial activity using biosynthesized AgNPs at 342 and 171 µg/mL and chemically synthesized at 235 and 117.5 µg/mL. Why change the concentration used?

7-    Please standardized the units such as hrs., hours, h.,

8-    Lines 576 – 577, “Turhan et al. (2022) and Klug et al. (2017) [55-56]” should be “Turhan et al. and Klug et al. [55-56]”, please check and revise throughout the manuscript.

9-    What about the pH effect on the chemical synthesis of AgNPs?

10- Figures legends should be rephrased and concise. The figure legend should not contain repeated data found in the text.

11- The title of Table 2 should be improved.

12- The concentrations used for antimicrobial activity are varied. Therefore, it may be the high activity of JAgNPs due to the high concentration, please clarify.

13- Please check the diameter of the clear zone in Table 2 (0.2, 1.6, 2.1,..). Did the authors neglect or subtract the diameter of the disk or what?

14- Line 312, please add the unit of clear zones.

15- The major issue in the current study is that the biological activity data has not undergone statistical analysis.

16- The SE or SD in Figures 11 and 12 is too high, why? Please clarify.

17- The manuscript should undergo English editing to correct the typo- and grammatical errors.

moderate English editing is needed 

Author Response

1-    The title should be improved to indicate the work, authors investigate antimicrobial and antibiofilm.

Response: Title was rewritten.

2-    The abstract should be rephrased to contain promising data for characterizations.

Response: The abstract was rephrased including characterization methods used

3-    “gram-negative” and “gram-positive” should be “Gram-negative” and “Gram-positive”, please revise throughout the manuscript.

Response: We have attended the suggestion

4- The introduction should be improved by referring to the importance of biosynthesized AgNPs using different biological entities followed by those synthesized by plant extract. I recommend to citing the following references: https://doi.org/10.3390/catal12050462; https://doi.org/10.3390/jof8040396.

We rewrite the introduction and add references suggested

5-    Line 531, the scientific names must be in italics, please check and revised throughout the manuscript.

Response: All scientific names in manuscript were italicized

6-    In line 526, the authors investigate the antimicrobial activity using biosynthesized AgNPs at 342 and 171 µg/mL and chemically synthesized at 235 and 117.5 µg/mL. Why change the concentration used?

Response: The use of different concentrations is clarified

7-    Please standardized the units such as hrs., hours, h.,

Response: All units such hours, minutes were standardized

8-    Lines 576 – 577, “Turhan et al. (2022) and Klug et al. (2017) [55-56]” should be “Turhan et al. and Klug et al. [55-56]”, please check and revise throughout the manuscript.

Response: The reference was corrected, all references were revised

9-    What about the pH effect on the chemical synthesis of AgNPs?

Response: It was cleared in materials and method section

10- Figures legends should be rephrased and concise. The figure legend should not contain repeated data found in the text.

Response: We rename figure legends

11- The title of Table 2 should be improved.

Response: It was modified

12- The concentrations used for antimicrobial activity are varied. Therefore, it may be the high activity of JAgNPs due to the high concentration, please clarify.

Response: We clarify in Materials and Results sections the different concentrations based on initial weight of nanoparticles

13- Please check the diameter of the clear zone in Table 2 (0.2, 1.6, 2.1,..). Did the authors neglect or subtract the diameter of the disk or what?

Response: It was cleared in materials and method section

14- Line 312, please add the unit of clear zones.

Response: Done

15- The major issue in the current study is that the biological activity data has not undergone statistical analysis.

Response: The statistical analysis in the diffusion test was complemented

16- The SE or SD in Figures 11 and 12 is too high, why? Please clarify.

Response: It is clarified that this may be due to the number of repetitions

17- The manuscript should undergo English editing to correct the typo- and grammatical errors.

Response: Typographic errors were corrected. We have made an English editing service from MDPI

Reviewer 3 Report

The manuscript entitled "Antibiofilm activity of chemically and biologically synthesized silver nanoparticlespresents the results of a study on the biosynthesize and characterization of silver nanoparticles using an aqueous extract of Jacaranda mimosifolia and chemically synthesized silver nanoparticles.  The authors evaluated the antibacterial and antibiofilm activities, as well as physicochemical properties. This is a well conducted study and the text is written concisely and consistently, and the data are sound. The Introduction explained why research was undertaken and the Abstract reflect all relevant aspects of the manuscript. The employed methods are appropriate and well described. The discussion and conclusions are well balanced and adequately supported by the properly presented data.

Author Response

Thank you for your comments. In addition, we have made an English editing service from MDPI.

Round 2

Reviewer 1 Report

All the Queries have been addressed and recommended for publication

Author Response

Thanks for your comments

Reviewer 2 Report

1- The authors return the uses of different concentrations due to the initial number of dry nanoparticles, this statement needs more clarification.

2- The reference style is not compatible (for instance, see line 601), please check and revised.

3- The scientific names must be in italics throughout the manuscript, in addition, reference section, please check and revised

4- Please add significant letters in Table 2

5- Some SE or SD in Figures 11 and 12 are too high, please clarify.

6- In the footnote of Figure 11, the authors should clarify is the significant difference between different MIC values for the same NPs or between two NPs at the same MIC value. this point should be addressed in other figures.

The manuscript needs moderate English editing

Author Response

1- The authors return the uses of different concentrations due to the initial number of dry nanoparticles, this statement needs more clarification.

Response: The use of different sample concentrations is explained in more detail

2- The reference style is not compatible (for instance, see line 601), please check and revised.

Response: Reference in line 601 was corrected (the year was deleted): We revised all document.

3- The scientific names must be in italics throughout the manuscript, in addition, reference section, please check and revised

Response: Scientific names were revised for italics form. We do not use italics when we refer to the common name of the plant (i.e. Jacaranda tree).

4- Please add significant letters in Table 2

Response: Table 2 was modified

5- Some SE or SD in Figures 11 and 12 are too high, please clarify.

We explain the variation due to the number of repetitions in lines 440 and 474

6- In the footnote of Figure 11, the authors should clarify is the significant difference between different MIC values for the same NPs or between two NPs at the same MIC value. this point should be addressed in other figures.

As in the previous observation, we explain de differences for Figure 11